# Biocomposites of Epoxidized Natural Rubber/Poly(lactic acid) Modified with Natural Fillers (Part I)

**DOI:** 10.3390/ijms22063150

**Published:** 2021-03-19

**Authors:** Anna Masek, Stefan Cichosz, Małgorzata Piotrowska

**Affiliations:** 1Faculty of Chemistry, Institute of Polymer and Dye Technology, Lodz University of Technology, Stefanowskiego 12/16, 90-924 Lodz, Poland; Stefan.cichosz@p.lodz.pl; 2Faculty of Biotechnology and Food Sciences, Institute of Fermentation Technology and Microbiology, Lodz University of Technology, Wolczanska 171/173, 90-924 Lodz, Poland; malgorzata.piotrowska@p.lodz.pl

**Keywords:** cellulose, poly(lactic acid), epoxidized natural rubber, polymer blend, biodegradation

## Abstract

The study aimed to prepare sustainable and degradable elastic blends of epoxidized natural rubber (ENR) with poly(lactic acid) (PLA) that were reinforced with flax fiber (FF) and montmorillonite (MMT), simultaneously filling the gap in the literature regarding the PLA-containing polymer blends filled with natural additives. The performed study reveals that FF incorporation into ENR/PLA blend may cause a significant improvement in tensile strength from (10 ± 1) MPa for the reference material to (19 ± 2) MPa for the fibers-filled blend. Additionally, it was found that MMT employment in the role of the filler might contribute to ENR/PLA plasticization and considerably promote the blend elongation up to 600%. This proves the successful creation of the unique and eco-friendly PLA-containing polymer blend exhibiting high elasticity. Moreover, thanks to the performed accelerated thermo-oxidative and ultraviolet (UV) aging, it was established that MMT incorporation may delay the degradation of ENR/PLA blends under the abovementioned conditions. Additionally, mold tests revealed that plant-derived fiber addition might highly enhance the ENR/PLA blend’s biodeterioration potential enabling faster and more efficient growth of microorganisms. Therefore, materials presented in this research may become competitive and eco-friendly alternatives to commonly utilized petro-based polymeric products.

## 1. Introduction

Poly(lactic acid) (PLA) is reported to be one of the most interesting biodegradable and renewable aliphatic polyesters [1,2,3]. Not only is it extensively investigated and widely utilized, but PLA also exhibits a great potential to replace commonly used petrochemical-based polymers for industrial applications, which is crucial from the environmental point of view.

In recent years, Hamad et al. [4] presented an interesting and broad review debating poly(lactic acid) blends. It covers some important and interesting aspects from PLA synthesis and large-scale production through PLA modification methods (including polymer blending) to possible applications, as well as future trends.

It is denoted that over the past years, noticeable attention has been drawn to the polymer blending technique as a way to modify PLA-based materials to obtain a product of desired properties [5,6,7,8,9,10,11,12,13]. Especially, rubber-incorporating blends are being intensely investigated. According to Hamad et al. [4], the first study that reported the creation of PLA/rubber blends was carried out by Ishida et al. [14].

In this work, various types of rubber were tested, e.g., nitryl–butadiene rubber (NBR), ethylene–acrylic rubber (AEM), ethylene–propylene rubber (EPM), isoprene rubber (IR) among their influence on the mechanical properties of PLA/rubber (90/10 wt%). The results obtained from the tensile tests showed that the blends prepared with the employment of NBR and IR exhibited elevated elongations, which was explained by the absence of the crosslinks within the rubber structure. Additionally, the NBR-incorporating blend was reported to possess the lowest particle size (3-4 µm) of the rubber domains, which, according to the research, resulted in improved impact strength. The impact strength in the case of this sample achieved a two-fold higher value compared to neat PLA.

In turn, the natural rubber (NR) amount’s (5, 10, 20 wt%) influence on the PLA/NR blend properties was studied by Bitinis et al. [15]. Scientists discovered that 10 wt% NR resulted in the highest PLA/NR blend elongation of approximately 200%. Yet, NR at 20 wt% caused a drop in the tensile properties. Authors attributed this phenomenon to the possible coalescence of rubber particles that occurred at higher NR content. Consequently, it led to a considerable average particle size rise and size distribution broadening.

Another interesting research was presented by Zhang et al. [16]. The authors investigated PLA, and epoxidized natural rubber (ENR) blends with 40 wt% of PLA content. The specimens were prepared with the use of an internal mixer in the presence of liquid NR (LNR) that played the role of a compatibilizer between the PLA and ENR. Tensile experiments revealed a noticeable improvement in the mechanical properties when the blend contained approximately 5 wt% of LNR. The elongation increase in the analyzed PLA/ENR samples was noted, and the authors concluded that LNR might have acted not only as a compatibilizer but also as a plasticizing agent. Then, the plasticization was confirmed with differential scanning calorimetry (DSC) as the reference PLA sample exhibited a Tg at 62.8 °C, but when LNR was added, its value decreased to approximately 58.8 °C.

A slightly different approach to PLA properties’ modification is to use the fillers of specific characteristics. Recently, natural substances are gaining significant attention [17,18,19,20,21,22] as they may contribute extensively to the creation of sustainable polymeric materials. Therefore, in the previous years, considerable interest has been given to the nature-derived additives and the effect of processing conditions on mechanical, rheological, and thermal properties of the PLA/natural filler composites [23,24,25,26,27,28,29,30,31,32]. 

It was shown by Barczewski et al. [33] that natural fillers may play an important role in epoxy-based polymer matrices. Additionally, Trifol et al. [34] examined nanoclay/nanocellulose PLA composites for food packaging, and Mysiukiewicz et al. [35] proved that natural substances may affect the degradation potential of PLA-based composites. Yet, even though some scientists published interesting results, less attention was drawn to PLA blend/natural additive systems. A few examples of works presenting filled PLA-based blends are presented below.

Boubekeur et al. [36] analyzed the effect of jute fibers on the properties of PLA and low-density polyethylene (LDPE) blends. The authors prepared samples consisting of 20 wt% LDPE, 80 wt% PLA, 5 per hundred rubber (phr) compatibilizer, and fibers (0–40 phr). It was found that the incorporation of jute fibers into PLA/LDPE blends contributed to the enhancement of the thermal stability and the modulus of the blend. 

Another interesting example is the work presented by Bocz et al. [37], who developed fully biodegradable blends of PLA and thermoplastic starch (TPS) filled with flax fibers. Scientists obtained a material that exhibited a synergic effect of high strength, stiffness, and thermal stability.

This research is a broadening of the works presented above. It debates the biodegradable and elastic ENR/PLA blends filled with different natural additives, which, until now, were only slightly described in the literature. The following work revealed some new aspects significant to this topic, namely, the influence of neat cellulose, flax fibers, and montmorillonite on the material properties, as well as aging of the blends, while subjected to different environmental conditions, e.g., elevated temperature, UV irradiation. Additionally, it was proven that plant-derived fibers incorporated into ENR/PLA blends might extensively enhance their biodegradation potential, enabling faster and more efficient growth of microorganisms. 

## 2. Results and Discussion

ENR/PLA blend samples were filled with commercially available cellulose (used commonly for paperboard production and paper coating), flax fibers, and montmorillonite. Plant-derived natural fibers were expected to enhance the biodegradation potential of the prepared samples and contribute to mechanical performance improvement. 

According to the literature, the reinforcement could be originated from probable interactions between the hydroxyl groups in cellulosic materials and oxirane rings of ENR/ester groups of PLA [38,39]. The scheme of possible interactions is shown in Figure 1. The aim of using various natural fibers (neat cellulose, flax fibers) was to investigate the possibility of flax fibers (also containing lignin and hemicellulose) employment instead of neat cellulose, which derivation includes some additional processes.

Additionally, montmorillonite (MMT) was employed in the role of a supplementary natural inorganic filler that could possibly affect the degradation characteristic and mechanical performance of the analyzed polymer blends.

As was mentioned above, samples prepared in this research were analyzed regarding the influence of the applied filler on the ENR/PLA blend properties and its potential impact on the specimen degradation process. Therefore, the carried-out measurements were carefully chosen. The static mechanical analysis was carried out to assess the influence of employed fillers on the performance of the prepared ENR/PLA blend specimens. Further, a swelling experiment was carried out to assess the crosslinking density as it was reported that the bonds created between different polymer macromolecules may have an influence on the mechanical properties of ENR/PLA blends and might exhibit a stabilizing effect on the properties of the material during the aging processes [40]. Moreover, it is inferred that the polymer composites of a higher polar component of surface free energy are more prone to degradation during the aging process regarding the ongoing radical reactions [41], e.g., oxidation. Therefore, the contact angle measurement, based on which the surface free energy value and its components were calculated, was carried out.

### 2.1. Characterization of Specimens Before the Accelerated Ageing Process

Regarding the data gathered in Figure 2a, it is possible to observe some variations in the swelling behaviour of prepared ENR/PLA polymer blend samples considering the filler type and employed crosslinking system. In turn, mass rise after the swelling process was proportional to the amount of solvent trapped in the structure of the analyzed specimen. The more solvent that was accommodated inside the polymer network, the lower the crosslinking degree between the polymer macromolecules. Thus, it could be concluded that natural fibers (cellulose fiber (CF), flax fiber (FF)) addition might contribute to the improvement in the density filler–polymer matrix interactions. This may be caused by both the physical interactions between hydroxyl groups of cellulosic material and oxirane rings of ENR or ester moieties of PLA (Figure 1) and covalent chemical bonds created during the vulcanization process. Thus, both physical and chemical crosslinking were present in the system described above. Moreover, lower swelling of CF- or FF-loaded specimens may also be affected by the filler’s poor affinity towards the solvent herein toluene.

On the contrary, MMT incorporation into ENR/PLA blends caused an opposite effect—the investigated sample was swollen similarly to the neat ENR/PLA blend. This may indicate that MMT particles, which are in the shape of plates, could behave as a steric blockade and prohibit the creation of crosslinks in the structure of the polymer blend. 

Moving forward, the mass loss detected during the swelling process might be attributed to some low molecular weight additives rinsed when the sample was put in the solvent environment. This could be explained by the fact that low molecular weight additives usually migrate to the surface of the polymeric material [42]. Therefore, washing out of some part of the applied additives is, in most cases, inevitable. Considering the errors accompanying the main bars (Figure 2a), the mass loss was on the same level throughout the analyzed samples, no matter what the filler was.

The highest tensile strength was evidenced in the case of the lowest swollen sample, namely ENR/PLA + FF. This results in the material stiffening and loss of the ability to elongate (Figure 2b,c). An intense improvement in tensile strength of the sample filled with FF in comparison to the CF-filled ENR/PLA blend could be caused by the different structures and performances of these two fillers. Flax fibers are composed of various additional substances rich in hydroxyl moieties, e.g., lignin, hemicellulose, which may affect the polymer matrix–filler interface and intensify the predicted interactions (Figure 1). On the other hand, cellulose is a material with a homogenous chemical composition, simultaneously being capable of establishing hydrogen bonds. Despite some similarities, cellulose and flax fibers may significantly differ considering their surface characteristics, e.g., hydrophilicity, specific surface area, accessibility of surface functional groups, which may affect the filler’s behavior in the polymer matrix and its effect on the properties of the polymer blend [43].

One should also consider the CF and FF behavior in the analyzed matrixes as natural fibers added to ENR/PLA blends may be concentrated only in a certain polymer domain and, thus, affect the blend properties in varied ways. According to the data gathered in the literature, cellulosic material itself may slightly contribute to the PLA tensile strength improvement—from approximately 65 MPa to 68 MPa (improvement of 5%) [44]. Yet, when cellulose was added to natural rubber, an increase in tensile strength from 12 MPa to almost 18 MPa (improvement of 50%) could be observed [45].

Moving forward, the sample filled with both FF and MMT exhibited lower tensile strength, which was on the level of the ENR/PLA + CF specimen. Nevertheless, the ENR/PLA + FF + MMT sample revealed the highest elongation at break of approximately 600%. This phenomenon could be explained with the synergic effect of MMT particles alignment within the structure of the polymer matrix (MMT plates are high aspect ratio particles; their orientation is a significant factor regarding the mechanical performance) and potentially lower crosslinking density (evidenced before), which leads to the material plasticization [46,47]. The effect of plasticization observed with MMT may be caused by free surfactant present in MMT.

A similar effect has been observed in work presented by Keawkumay et al. [48]. According to the results presented in the research, a certain MMT surface treatment may contribute to the lowering of rubber’s crosslinking density and, thus, decrease the mechanical performance of a composite. 

Moreover, Wang et al. [49] underlined the importance of appropriate filler dispersion within the polymer matrix. The authors presented results for MMT contents from 1–5 wt%. A significant increase in the composite tensile strength was only observed in the case of low MMT contents. 

On the other hand, Jiang et al. [50] drew attention to the problem of high specific surface area and shape of MMT particles—with their large L/D ratio, the MMT platelets induced lower stress concentration, which partly contributed to the higher elongation of MMT-filled blend. Similarly, scientists report that the highest values of tensile strength and elongation at break are evidenced for the MMT content up to 2.5 wt%. This indicates that a specific aluminosilicate amount is required to ensure the most satisfying mechanical properties of a final product.

Additionally, it was proven by Papageorgiou et al. [51] and Mathew et al. [52] that natural additives, such as MMT and cellulose, might significantly influence the crystallization behavior contributing to some variations in composite mechanical properties. According to the studies, these fillers may behave as nucleating factors and promote the crystalline domain’s formation. Yet, it was reported that MMT incorporation may result in the considerable imperfectness of created crystals, and, thus, it might contribute to the lowering of the stiffness and performance of a PLA-containing composite.

Confirmation for the analyzed ENR/PLA blend stiffening and plasticization can be found in Table 1, where the tensile tension values at elongation of 100%, 200%, and 300% are presented. It is visible that the specimen filled with FF achieved the tension of (16.2 ± 0.4) MPa at the elongation of only 100%, confirming its high stiffness. Additionally, the lowest value of tensile tension at 300% was attributed to the sample filled with both FF and MMT, which was supposed to be plasticized. 

Finally, the surface free energy (SFE) and its components are presented in Figure 2d. It may be observed that the incorporation of cellulose and flax fibers into the ENR/PLA blend did not significantly affect the surface energy characteristics. Yet, the sample filled with both FF and MMT exhibited lower total SFE and relatively slight polar component.

This phenomenon is also visible regarding the data gathered in Table 2. ENR/PLA reference blend and specimens filled with cellulose either flax fibers revealed similar water and diiodomethane droplets’ behavior on the surface of the polymer blend. Nonetheless, the MMT-containing sample exhibited the contact angle for water of (97 ± 2)° and in the case of diiodomethane-(71 ± 2)°, which were significantly higher values, when compared to the rest of the specimens.

It is often inferred that materials exhibiting a higher polar part of surface free energy (SFE) could be more affected by the aging processes [41]. Therefore, taking into consideration data gathered in Figure 2d and Table 2, ENR/PLA blend filled with both FF and MMT could exhibit improved resistance to the aging processes. Yet, as FF and MMT are natural additives, the blend should still be prone to the fungi environment similar to the specimens filled only with plant-derived fibers, which exhibit a higher polar part component of surface free energy.

### 2.2. Characterization of the Aging Impact

Polymer blend samples were subjected to an accelerated aging process to assess their resistance to material degradation induced by temperature (thermo-oxidative aging) and UV light (UV irradiation; 340 nm). Both aging processes indicated radical reactions, which may contribute to the oxidation, crosslinking or chain scission within the polymeric material [53]. The progress of the degradation processes was tracked via the mass control, color change, and mechanical properties variations over accelerated aging time, which was 200 h. Time intervals between the measurements were established for 50 h (from 0–200 h) as the prepared samples significantly degraded during the relatively short period, 5 h.

A summary of experiments carried out can be found in Figure 3. In general, the mass loss among the performed aging processes (Figure 3a) was higher for thermo-oxidative aging, which could be connected with the temperature-induced moisture evaporation from the natural fibers employed as fillers in the investigated polymer blends. The moisture content in plant-derived fibers is not homogenous and repetitive [54]. Therefore, the mass loss processes might be less regular among the investigated samples for thermo-oxidative aging in comparison with UV aging. 

Considering Figure 3b, the specimens changed their visual features during both aging processes in varied ways. Thermo-oxidation caused a relatively high and quick color change in the PLA/ENR blend filled with both FF and MMT in comparison with the UV aging during which the color variations’ rate was steadier. On the other hand, it could be said that UV irradiation caused more significant changes for different samples after 200 h of the aging process.

What should be underlined is samples of reference PLA/ENR blends became too brittle after just 100 h of thermo-oxidation/UV irradiation. Thus, it was impossible to carry out the color change measurements after this time as it was necessary to press the sample against the measuring instrument. 

Additionally, variations of some additional parameters, such as whiteness index, chroma, and hue angle, which contribute to the final color change, are presented in Table 3. Regarding the gathered data, it might be concluded that UV aging leads to the fading of specimens resulting in less intense color visible for the human eye. Contrary, thermo-oxidative aging results in more complex and less predictable variations.

Considering the changes in the mechanical properties of the analyzed polymer blends, it could be observed that regarding the thermo-oxidative (Figure 3c,d) and UV (Figure 3e,f) aging, samples filled with flax fibers exhibited higher resistance to the aging processes in comparison with the reference ENR/PLA blend or CF-filled ENR/PLA specimen. Moreover, as was mentioned before, prepared ENR/PLA blends seemed to be more resistant to UV irradiation, e.g., samples filled with CF and FF were not degrading that fast when subjected to UV aging (the tensile tests were possible to carry out; samples were flexible enough to enable the measurement).

Moving forward and comparing aging processes, it was found that UV irradiation usually caused temporary tensile strength improvement, which indicates that UV irradiation may favor the crosslinking processes, e.g., preliminary tensile strength rise for ENR/PLA + CF or ENR/PLA + FF during the UV aging. On the contrary, the increased temperature probably led to the material’s intense oxidation and polymer chain scission.

As expected, the performed accelerated aging revealed the high aging resistance of the sample filled with both FF and MMT. This could be explained with the lower polar part component of the surface free energy of the analyzed specimen. Yet, barrier characteristics of MMT-filled materials and the presence of an aging-resistant inorganic fraction ought to be considered [55]. It was reported that MMT might contribute extensively to the reduction in oxygen permeability to the interior of the composite and, thus, prohibit the aging process (oxidation cannot occur) [56,57]. 

In Table 4, the aging coefficient values, calculated based on ENR/PLA blend tensile strength and elongation at break before and after the performed aging processes, are presented. The lower the *K* value, the more degraded the material. According to the gathered data, polymer blend samples filled only with plant-derived fibers were almost fully degraded after only 50 h of performed accelerated aging processes. Nonetheless, the incorporation of an inorganic faction into ENR/PLA blends may successfully lead to the increased aging-resistance of the sample, e.g., when aging lasts for 200 h most of the samples were fully degraded, and it was impossible to carry out the measurement, but when MMT was incorporated, the measurement was possible to conduct and the blend’s mechanical properties’ loss was on the level of 80% (*K* was approximately 0.2).

### 2.3. Microorganism Growth Tests

In the previous subsection, it was proved that the ENR/PLA blend degradation rate in a thermo-oxidative environment and under UV irradiation may be controlled with different natural additives, such as natural fibers or aluminosilicates. MMT incorporation contributed extensively to the improvement in the aging resistance of the investigated polymer blends.

The materials analyzed in this research were also subjected to some tests regarding the resistance of ENR/PLA blends filled with natural additives to molds. Data gathered from this experiment is presented in Table 5. It is visible that the natural fiber incorporation into the polymer matrix improved the microorganisms’ growth and impaired the material ability to resist molds.

Moreover, MMT presence in the ENR/PLA blend, which was the reason for the delayed material degradation when subjected to increased temperature or UV irradiation, did not affect the molds’ growth on the surface of the investigated polymeric material.

According to the presented molecular growth evaluation (MGE) and information given in the standard, none of the analyzed materials was resistant to microorganism growth. Nonetheless, filling ENR/PLA blends with commercially available neat cellulose fibers and flax fibers may only increase the degradation potential of prepared materials. What is also important is inorganic silicates, which may improve the mechanical properties of polymer blends and contribute to the prolonged time of use in certain conditions, did not have a negative effect on the final product biodegradation.

## 3. Materials and Methods

### 3.1. Materials

The subject of the study, epoxidized natural rubber (ENR) (Epoxyprene ENR-50; 50 mol% epoxidation), was obtained from Kumpulan Guthrie Berhad (Kuala Lumpur, Malaysia). Poly(lactic acid) (PLA) was provided by Nature Works (Minnetonka, Minnesota, USA). Cellulose fibers (CF) employed in this research were the Arbocel UFC100 Ultrafine Cellulose for Paper and Board Coating (UFC100) from J. Rettenmaier & Soehne (Rosenberg, Germany). It was in powder form (density: 1.3 g/cm^3^, average fiber length: 6–12 µm, pH: 5.0–7.5). Flax fibers (FF) employed in this research were supplied by Łukasiewicz Research Network—Institute of Biopolymers and Chemical Fibers (Lodz, Poland). Fibers were comminuted and composed of lignin, hemicellulose, and cellulose in an unknown ratio. Montmorillonite (MMT) modified nanoclay was purchased from Sigma–Aldrich (Darmstadt, Germany). It contained 25–30 wt% of octadecylamine grafted onto the surface. MMT particle size was approximately 20 µm. Lauric acid (97% purity), 1,2-dimethylimidazole (highest available purity), obtained from Sigma–Aldrich (Darmstadt, Germany), and elastin hydrolysate, purchased from Proteina (Lodz, Poland), were used as a crosslinking system.

### 3.2. Preparation of ENR/PLA Blend Samples

Natural fibers and poly(lactic acid) were dried for 24 h at, respectively, 100 °C and 70 °C in a laboratory oven (Binder, Tuttlingen, Germany) before being incorporated into the polymer matrix. All mixture components (Table 6) were put into a micromixer (Brabender Lab-Station from Plasti-Corder with Julabo cooling system) at 160 °C for 30 min (50 rpm). Next, such prepared material was placed between two roll mills with 100 × 200 mm rolls, at a roll’s temperature of 20–25 °C and friction of 1:1.1 for approximately 2 min. The last step was to compress the polymer blend plates in a hydraulic press at 160 °C (electrically heated platens) for 60 min at approximately 125 bar. Therefore, the mixture was put between two steel molds and Teflon sheets.

### 3.3. Accelerated Ageing of the Materials

#### 3.3.1. Thermo-Oxidative Ageing

Thermo-oxidative aging was performed in a laboratory oven (Binder, Tuttlingen, Germany) at 70 °C for 200 h. The measurement was aimed at examining the degradation progress of the samples over time. For this purpose, 4 samples of each ENR/PLA blend were placed in an oven and were taken out individually after, respectively, 50 h, 100 h, 150 h, and 200 h.

#### 3.3.2. UV irradiation

ENR/PLA blend samples were put in special metal folders and placed in the Atlas UV 2000 apparatus (Duisburg, Germany). Again, the measurement was aimed at examining the degradation progress of the samples over time. For this purpose, specimens were taken out individually after, respectively, 50 h, 100 h, 150 h, and 200 h. The aging cycle consisted of two alternating segments: day segment (240 min, 60 °C, UV irradiation: 0.7 W/m^2^) and night segment (120 min, 50 °C, no UV radiation).

### 3.4. Methods of Polymer Blend Sample Characterization

#### 3.4.1. Swelling in Toluene

The principle of the measurement was based on measuring the weight gain of the sample that was subjected to the organic solvent environment (toluene). From each vulcanizate, 4 samples of different shapes were cut out (the weight of each ranged from 30 to 40 mg). The samples were weighed before the measurement (m_1_) and then immersed in a solvent. After 48 h, they were taken out and weighed again (m_2_). Before weighing the wet sample, the excess toluene was cleaned with filter paper, and then the sample was immersed in ethyl ether for 1−2 s. The samples were dried to constant weight in an oven at 50 °C for 96 h and weighed again (m_3_). To analyze the potential crosslinking density, two parameters were calculated: m_rise_ = m_2_ − m_1_ (m_rise_ reveals how much the sample is swollen in the solvent), m_loss_ = m_3_ − m_1_ (m_loss_ informs about the low molecular weight compounds washed out during the swelling in toluene).

#### 3.4.2. Contact Angle Measurement

Surface free energy was determined based on contact angle measurements done for two liquids: distilled water, diiodomethane (droplet volume of 1 μL). Before the measurement, the blend’s surface was cleaned with acetone. An OCA 15EC goniometer by DataPhysics Instruments GmbH (Filderstadt, Germany) equipped with a single direct dosing system (0.01–1 mL B. Braun syringe, Hassen, Germany) was employed, and the surface free energy was calculated thanks to the Owens–Wendt–Rabel–Kaelble (OWRK) method.

#### 3.4.3. Tensile Tests

Mechanical properties, such as tensile strength (TS) and elongation at break (Eb), were determined with the use of a Zwick-Roell 1435 measuring device (Ulm, Germany). Tests were carried out on a “dumbbell” shape, approximately 1.5 mm thick and 4 mm width specimens, according to PN-ISO 37:1998 standard. Specimens were cut out with the use of a punch described in the standard. Based on the obtained results, the aging coefficient *K* was calculated as a quotient of the product of TS and Eb after and before the performed aging process [58].

#### 3.4.4. Color Change

Optical properties’ characterization was determined with a Spectrophotometer UV-VIS CM-36001 from Konica Minolta. Sample color was described with the CIE-Lab system (*L*—lightness, *a*—red-green, *b*—yellow-blue). Then, according to the equations given below (1–4), color difference (Δ*E*), whiteness index (*W_i_*), chroma (*C_ab_*), and hue angle (*h_ab_*) values were calculated. Δ*a*, Δ*b*, Δ*L* were the coordinate differences between aged and unaged samples:(1)$$\Delta E=\sqrt{{\left(\Delta a\right)}^{2}+{\left(\Delta b\right)}^{2}+{\left(\Delta L\right)}^{2}}$$
(2)$$W_{i}=100-\sqrt{a^{2}+b^{2}+{\left(100-L\right)}^{2}}$$
(3)$$C_{ab}=\sqrt{a^{2}+b^{2}}$$
(4)$$h_{ab}=\left\{\begin{array}{c} arctg\left(\frac{b}{a}\right),\;when\;a>0\wedge b>0\\ 180{}^{\circ}+arctg\left(\frac{b}{a}\right),\;when\;\left(a<0\wedge b>0\right)\vee (a<0\wedge b<0)\\ 360{}^{\circ}+arctg\left(\frac{b}{a}\right),\;when\;a>0\wedge b<0 \end{array}\right.$$

#### 3.4.5. Mass Loss during the Degradation Process

Before and during the accelerated aging process, specially prepared samples were weighed at the following times: 0 h, 50 h, 100 h, 150 h, and 200 h. The experiment aimed to show how the mass of the specimens subjected to accelerated aging changed during the degradation.

### 3.5. Microorganism Growth Tests

The effect of molds on the materials tested was assessed in accordance with ISO 846. A mixed suspension in salt solutions of *Aspergillus niger*, *Paecilomyces variotii*, *Chaetomium globosum*, *Trichoderma viride,* and *Penicillium funiculosum* strains was used as a test organism. In tests, two microbial media were used: (1) incomplete nutrient solution medium (NaNO_3_ 2.0 g; KH_2_PO_4_ 0.7 g; K_2_HPO_4_ 0.3 g; KCl 0.5 g; MgSO_4_ 7H_2_O 0.01 g, agar 20 g; water 1000 mL, pH 6.0–6.5) for method A, and (2) complete nutrient medium with the composition as above with 30 g of glucose for method B and variant B’.

In method A, the natural resistance of the material when no other nutrient substance was present was tested. The material was disinfected with 75% ethanol and then placed on medium (1), and a suspension of molds was added (batch I). Simultaneously, batch S was prepared without microorganisms as a control. 

In method B, the fungistatic effect and the influence of surface contamination on the resistance of the material were tested. Material samples without disinfection were placed on medium (2), followed by a uniform deposition of the microorganism suspension. In method B’, first, the mold suspension was plated on the medium, and after 5 days of incubation at 25 °C, the test material was placed on a plate surface.

The samples were incubated at 28 °C and relative humidity (RH) 80% for 5 weeks (Climatic Chamber BINDER).

Afterward, the intensity of growth of microorganisms on both the nutrient media and samples was observed and evaluated using an assessment scale from 0 to 5 in accordance with ISO 846: no visible growth under the microscope (0); visible under a microscope (1); growth covering up to 25% of the sample area (2); growth covering up to 50% of the sample area (3); significant growth, covering more than 50% of the sample area (4); intensive growth covering the entire surface of the sample (5).

The final interpretation was for method A: 0—the material is not a breeding ground for microorganisms; 1—the material contains nutrients or is slightly contaminated to allow slight growth; 2–5—the material is not resistant to the action of microorganisms and contains substances that support their growth. For method B/B’: 0—strong fungistatic effect; 1–5—lack of fungistatic effect.

## 4. Conclusions

Concluding, the elastic blends of epoxidized natural rubber (ENR) with poly(lactic acid) (PLA) were successfully created. Regarding the fact that they were reinforced with flax fiber (FF) and montmorillonite (MMT), some additional functional properties were introduced, e.g., improved biodegradation potential, improved mechanical performance/elongation.

Thanks to the research carried out, it was shown that FF incorporation into ENR/PLA blend caused a significant improvement in tensile strength up to (19 ± 2) MPa, simultaneously contributing to the loss of an ability of a polymer blend to elongate (material stiffening). On the contrary, MMT employment in the role of the filler led to ENR/PLA blend plasticization and enabled the polymer blend’s elongation on the level of approximately 600%. Nevertheless, MMT-filled material exhibited significantly lower tensile strength in comparison with the samples filled with only plant-derived fibers.

Moreover, based on the gathered results, it can be concluded that the composite samples subjected to elevated temperature and UV irradiation underwent some notable changes in their properties. It was established that MMT incorporation delayed the degradation of the ENR/PLA elastic blends under the abovementioned conditions without affecting their biodeterioration potential. Thanks to the mold tests carried out, it was proven that plant-derived fibers incorporated into ENR/PLA blends significantly enhanced their biodegradation potential, enabling faster and more efficient growth of the microorganisms. Importantly, the biodeterioration ability was not affected by the MMT incorporation.

Significantly, it should be mentioned that, in comparison with literature data, ENR/PLA blends prepared in this research not only did exhibit improved degradation potential but were also characterized by a performance higher than noted for neat ENR vulcanizates. According to the article by Intharapat et al. [59], unfilled ENR products reveal a tensile strength of approximately 2 MPa. Additionally, when ENR is filled with rice husk ash, the vulcanizate’s performance might be elevated up to 16 MPa [60], which is lower than the tensile strength of FF-filled ENR/PLA blend (approximately 19 MPa). Obviously, the ENR/PLA blends’ ability to elongate was limited by the PLA content, and, thus, the elongation at break of neat ENR vulcanizates was usually higher.

Gathering the presented information, it can be stated that biodegradable and elastic ENR/PLA blends are good materials for producing consumer products characterized by a short or long lifespan depending on the organic–inorganic phase ratio, which can be controlled by the amounts of different natural additives. Furthermore, the pro-ecological character of the produced polymer blends and their mechanical performance make them competitive and enable their widespread application in various branches of industry, while meeting current ecological requirements.

## Figures and Tables

**Figure 1 ijms-22-03150-f001:**
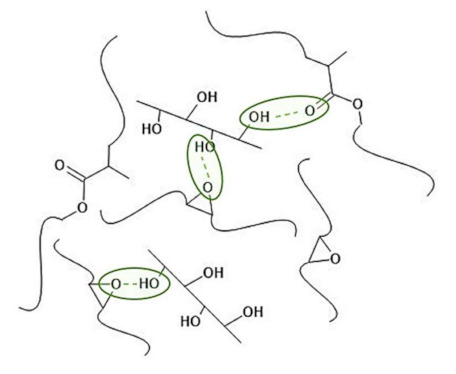
Examples of possible interaction in epoxidized natural rubber/poly(lactic acid) (ENR/PLA) blends between the hydroxyl group of cellulose and oxirane ring of ENR or ester group of PLA.

**Figure 2 ijms-22-03150-f002:**
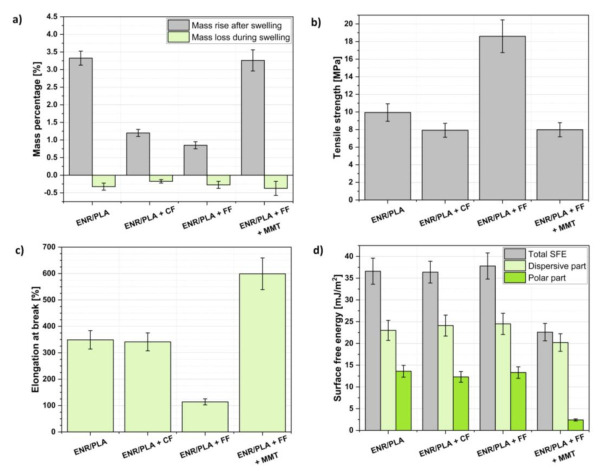
ENR/PLA blend properties before the accelerated aging processes: (**a**) analysis of the swelling experiment, (**b**) tensile strength of analyzed specimens, (**c**) elongation at break of investigated samples, (**d**) surface free energy, and its components analysis.

**Figure 3 ijms-22-03150-f003:**
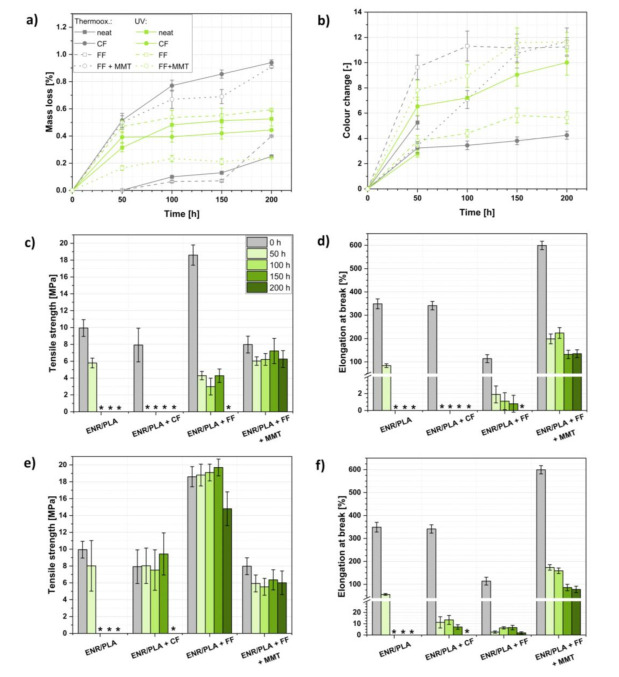
Investigation of the ENR/PLA blends’ properties before and after the accelerated aging processes: (**a**) mass loss during thermo-oxidative and UV aging, (**b**) color change during thermo-oxidative and UV aging, (**c**) tensile strength changes and (**d**) elongation at break variations during thermo-oxidative aging, (**e**) tensile strength changes and (**f**) elongation at break variations during the UV aging. * sample too brittle to be examined with the selected method; the measurement was impossible.

**Table 1 ijms-22-03150-t001:** Tensile tension at an elongation of 100%, 200%, 300% for the unfilled and filled epoxidized natural rubber/poly(lactic acid) (ENR/PLA) blend samples.

Sample	Tensile Tension [MPa] at Elongation of:		
	100%	200%	300%
ENR/PLA	4.5 ± 0.2	6.7 ± 0.3	8.8 ± 0.2
ENR/PLA + CF	5.9 ± 0.2	6.6 ± 0.4	7.7 ± 0.5
ENR/PLA + FF	16.2 ± 0.4	-----	-----
ENR/PLA + FF + MMT	5.0 ± 0.5	6.1 ± 0.2	6.6 ± 0.3

**Table 2 ijms-22-03150-t002:** Water and diiodomethane contact angles for the unfilled and filled ENR/PLA blend samples.

Sample	Contact Angle [°]	
	Water	Diiodomethane
ENR/PLA	70 ± 3	57 ± 2
ENR/PLA + CF	71 ± 3	56 ± 2
ENR/PLA + FF	69 ± 1	55 ± 3
ENR/PLA + FF + MMT	97 ± 2	71 ± 2

**Table 3 ijms-22-03150-t003:** Additional parameters contributing to the color change in ENR/PLA blends and their variations during the thermo-oxidative and UV accelerated aging processes: whiteness index, chroma, hue angle. * sample too brittle to be examined with the selected method; the measurement was impossible.

**Sample**	**Whiteness index Wi [-] – thermooxidation**				
	**0 h**	**50 h**	**100 h**	**150 h**	**200 h**
ENR/PLA	40 ± 4	37 ± 1	*	*	*
ENR/PLA + CF	41 ± 3	38 ± 5	39 ± 2	38 ± 3	38 ± 2
ENR/PLA + FF	41 ± 2	37 ± 3	36 ± 2	35 ± 4	39 ± 2
ENR/PLA + FF + MMT	46 ± 2	48 ± 2	41 ± 4	40 ± 1	40 ± 3
**Sample**	**Whiteness index Wi [-] – UV irradiation**				
	**0 h**	**50 h**	**100 h**	**150 h**	**200 h**
ENR/PLA	40 ± 4	42 ± 2	*	*	*
ENR/PLA + CF	41 ± 3	40 ± 3	42 ± 1	41 ± 2	48 ± 4
ENR/PLA + FF	41 ± 2	41 ± 2	42 ± 2	40 ± 2	42 ± 2
ENR/PLA + FF + MMT	46 ± 2	42 ± 3	41 ± 2	40 ± 3	45 ± 2
**Sample**	**Chroma Cab [-] – thermooxidation**				
	**0 h**	**50 h**	**100 h**	**150 h**	**200 h**
ENR/PLA	45 ± 2	50 ± 4	*	*	*
ENR/PLA + CF	44 ± 2	46 ± 2	44 ± 1	47 ± 2	48 ± 4
ENR/PLA + FF	47 ± 3	45 ± 3	45 ± 3	47 ± 2	46 ± 3
ENR/PLA + FF + MMT	43 ± 2	44 ± 1	47 ± 3	42 ± 2	46 ± 2
**Sample**	**Chroma Cab [-] – UV irradiation**				
	**0 h**	**50 h**	**100 h**	**150 h**	**200 h**
ENR/PLA	45 ± 2	46 ± 2	*	*	*
ENR/PLA + CF	44 ± 2	39 ± 3	43 ± 2	36 ± 2	36 ± 2
ENR/PLA + FF	47 ± 3	42 ± 2	43 ± 3	40 ± 1	41 ± 2
ENR/PLA + FF + MMT	43 ± 2	43 ± 3	43 ± 2	40 ± 3	40 ± 1
**Sample**	**Hue angle h_ab_ [°] – thermooxidation**				
	**0 h**	**50 h**	**100 h**	**150 h**	**200 h**
ENR/PLA	83 ± 4	76 ± 2	*	*	*
ENR/PLA + CF	75 ± 3	73 ± 4	73 ± 3	74 ± 4	73 ± 3
ENR/PLA + FF	81 ± 3	76 ± 3	75 ± 2	74 ± 2	77 ± 2
ENR/PLA + FF + MMT	82 ± 2	81 ± 2	77 ± 3	73 ± 3	74 ± 2
**Sample**	**Hue angle hab [°] – thermooxidation**				
	**0 h**	**50 h**	**100 h**	**150 h**	**200 h**
ENR/PLA	83 ± 4	78 ± 3	*	*	*
ENR/PLA + CF	75 ± 3	74 ± 3	75 ± 2	74 ± 3	71 ± 2
ENR/PLA + FF	81 ± 3	78 ± 3	79 ± 2	79 ± 1	79 ± 2
ENR/PLA + FF + MMT	82 ± 2	77 ± 2	75 ± 2	74 ± 3	77 ± 2

**Table 4 ijms-22-03150-t004:** Aging coefficients attributed to the filled and unfilled ENR/PLA specimens at a certain aging time. * sample too brittle to be examined with the selected method; the measurement was impossible.

**Sample**	**Aging coefficient K [-]—thermo-oxidation**			
	**50 h**	**100 h**	**150 h**	**200 h**
ENR/PLA	0.14 ± 0.06	*	*	*
ENR/PLA + CF	*	*	*	*
ENR/PLA + FF	0.004 ± 0.002	0.002 ± 0.001	0.002 ± 0.001	*
ENR/PLA + FF + MMT	0.3 ± 0.1	0.3 ± 0.1	0.20 ± 0,08	0.20 ± 0,07
**Sample**	**Ageing coefficient K [-]—UV irradiation**			
	**50 h**	**100 h**	**150 h**	**200 h**
ENR/PLA	0.13 ± 0.05	*	*	*
ENR/PLA + CF	0.03 ± 0.01	0.06 ± 0.03	0.02 ± 0.01	*
ENR/PLA + FF	0.02 ± 0.01	0.06 ± 0.02	0.06 ± 0.02	0.01 ± 0.01
ENR/PLA + FF + MMT	0.2 ± 0.1	0.18 ± 0.07	0.15 ± 0.05	0.21 ± 0.08

**Table 5 ijms-22-03150-t005:** The effect of molds on the unaged ENR/PLA blends; MGE—microorganism’s growth evaluation.

Sample	Resistance to Moulds	Fungistatic Effect
Method A	Method B	Method B’
MGE [-]	Picture	MGE [-]	Picture	MGE [-]	Picture
ENR/PLA	2	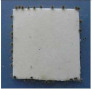	4	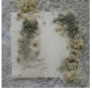	3	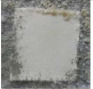
ENR/PLA + CF	3	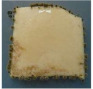	5	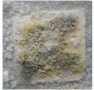	5	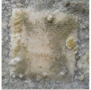
ENR/PLA + FF	5	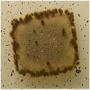	4	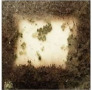	3	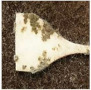
ENR/PLA + FF + MMT	5	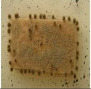	4	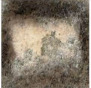	3	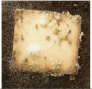

**Table 6 ijms-22-03150-t006:** Composition of the polymer blend mixtures prepared for analysis in this research. Abbreviations: ENR—epoxidized natural rubber, PLA—poly(lactic acid), LA—lauric acid, DMI—1,2-dimethylimidazole, EH—elastin hydrolysate, CF—cellulose fibers, FF—flax fibers, MMT—montmorillonite, phr—per hundred rubber (it means: for one hundred parts by weight of rubber there are x parts by weight of the substance).

Sample	Polymer Mixture Composition [phr]							
	ENR	PLA	LA	DMI	EH	CF	FF	MMT
ENR/PLA	100	75	3	0.6	0.6	----	----	----
ENR/PLA + CF	100	75	3	0.6	0.6	25	----	----
ENR/PLA + FF	100	75	3	0.6	0.6	----	25	----
ENR/PLA + FF + MMT	100	75	3	0.6	0.6	----	12.5	12.5

## Data Availability

No data available.

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
