# Peer review of "Biocomposites of Epoxidized Natural Rubber/Poly(lactic acid) Modified with Natural Fillers (Part I)"

_ijms, 2021, doi:10.3390/ijms22063150_

Round 1
Reviewer 1 Report
This paper deals with the preparation and characterization of vulcanized ENR/PLA blends reinforced with cellulose, flax fiber and montmorillonite. Tensile properties, toluene swelling behavior, and surface free energy were determined. Moreover, the samples were exposed to accelerated thermo-oxidative and UV ageing and test of mould growth.
The topic is interesting, but in my opinion in a few points it is not clear the role of each component of the composite materials with respect to the measured property. Please, consider the following points.
- Pags 3-4. The authors found that the addition of CF and FF to ENR/PLA causes a decrease of the amount of solvent entrapped into the material and they explain this result by considering the interactions between the OH groups of the fillers and the functional moieties of both ENR and PLA. Perhaps, this effect could be demonstrated by FTIR analysis and by solvent extraction.
- At pag. 4 lines 144-146 the authors wrote that intercalated MMT particles could hinder the crosslinking reaction between polymer chains. This behavior is not clear and in addition the authors did not demonstrate that the morphology of the composite is intercalated.
- The authors affirm that the tensile strength of composites filled with FF and CF is correlated to the different structures of the two fillers. However, no details are given about the dimension of FF and about their morphology.
- The effect of plasticization observed with MMT may be caused by free surfactant present in MMT.
Author Response
Institute of Polymer and Dye Technology
Lodz University of Technology
90-924 Lodz, ul. Stefanowskiego 12/16, Poland
Tel.: +48 42 631 32 93, Fax: +48 42 636 25 43
March 8, 2021
International Journal of Molecular Science MDPI
Dear Editor,
I am sending you our manuscript entitled “Biocomposites of epoxidized natural rubber/ polylactic acid (ENR/PLA) modified by natural filler” by Anna Masek, Stefan Cichosz and Malgorzata Piotrowska with a request to publish the paper in International Journal of Molecular Science MDPI. The manuscript has not been previously published, is not currently submitted for review to any other journal, and will not be submitted elsewhere before a decision is made by this journal. All authors have approved the manuscript and agree with its submission to International Journal of Molecular Science MDPI.
For correspondence please use the following information: corresponding author: Anna Masek
Institute of Polymer and Dye Technology Technical University of Lodz
90-924 Lodz, ul. Stefanowskiego 12/16, Poland
Tel.: +48 42 631 32 13
Fax.: +48 42 636 25 43
e-mail: anna.masek@p.lodz.pl
PhD, DSc, Anna Masek
Associate Professor
Technical University of Lodz, Institute of Polymer and Dye Technology,
Stefanowskiego 12/16, 90-924 Lodz, Poland
Reviewer #1
Thank you for reviewing the submitted work. Thank you for your positive and critical words about our manuscript.
This paper deals with the preparation and characterization of vulcanized ENR/PLA blends reinforced with cellulose, flax fiber and montmorillonite. Tensile properties, toluene swelling behavior, and surface free energy were determined. Moreover, the samples were exposed to accelerated thermo-oxidative and UV ageing and test of mould growth.
The topic is interesting, but in my opinion in a few points it is not clear the role of each component of the composite materials with respect to the measured property. Please, consider the following points.
- Pags 3-4. The authors found that the addition of CF and FF to ENR/PLA causes a decrease of the amount of solvent entrapped into the material and they explain this result by considering the interactions between the OH groups of the fillers and the functional moieties of both ENR and PLA. Perhaps, this effect could be demonstrated by FTIR analysis and by solvent extraction.
- Answer: Thank you for your attention, of course, we fully agree with the Reviewer 1, but due to the lack of research material, we will publish the detailed results based on the FTIR analysis in the next paper.
- At pag. 4 lines 144-146 the authors wrote that intercalated MMT particles could hinder the crosslinking reaction between polymer chains. This behavior is not clear and in addition the authors did not demonstrate that the morphology of the composite is intercalated.
Answer: Thank you for your valuable attention, the comment has been deleted. We have generally described the morphology (plate shape) of MMT on the basis of a lot of literature data.
- The authors affirm that the tensile strength of composites filled with FF and CF is correlated to the different structures of the two fillers. However, no details are given about the dimension of FF and about their morphology.
- Answer: Unfortunately, we no longer have research material or funds available for further research on the dimension of FF and about their morphology. However, we will try to implement this remark in the next document.
- The effect of plasticization observed with MMT may be caused by free surfactant present in MMT
- Answer: Thank you for Your important attention, the request was included in the description.
Reviewer 2 Report
The work is basically well done and properly described but I’m not convinced that it has anything enough originality.
The aim of the study was to prepare degradable elastic blends of epoxidized natural rubber with poly(lactic acid) that were reinforced with flax fiber and montmorillonite. While the International Journal of Molecular Sciences provides an advanced forum for molecular studies in biology and chemistry, with a strong emphasis on molecular biology and molecular medicine.
According to the authors this article filling the gap in the literature regarding the PLA-containing polymer blends filled with natural additives. In world literature is a lot report and research concerning on PLA-containing polymer blends filled with natural additives.
This manuscript isn't suitable for this journal, becouse the aim IJMS is to provide publication of cutting-edge research for molecular studies in biology and chemistry, with a strong emphasis on molecular biology and molecular medicine.
Author Response
Institute of Polymer and Dye Technology
Lodz University of Technology
90-924 Lodz, ul. Stefanowskiego 12/16, Poland
Tel.: +48 42 631 32 93, Fax: +48 42 636 25 43
March 8, 2021
International Journal of Molecular Science MDPI
Dear Editor,
I am sending you our manuscript entitled “Biocomposites of epoxidized natural rubber/ polylactic acid (ENR/PLA) modified by natural filler” by Anna Masek, Stefan Cichosz and Malgorzata Piotrowska with a request to publish the paper in International Journal of Molecular Science MDPI. The manuscript has not been previously published, is not currently submitted for review to any other journal, and will not be submitted elsewhere before a decision is made by this journal. All authors have approved the manuscript and agree with its submission to International Journal of Molecular Science MDPI.
For correspondence please use the following information: corresponding author: Anna Masek
Institute of Polymer and Dye Technology Technical University of Lodz
90-924 Lodz, ul. Stefanowskiego 12/16, Poland
Tel.: +48 42 631 32 13
Fax.: +48 42 636 25 43
e-mail: anna.masek@p.lodz.pl
PhD, DSc, Anna Masek
Associate Professor
Technical University of Lodz, Institute of Polymer and Dye Technology,
Stefanowskiego 12/16, 90-924 Lodz, Poland
Reviewer #2
Thank You for reviewing the submitted work. Thank you for your positive and critical words about our manuscript. Our research publication, which we have been publishing for a long time in the International Journal of Molecular Science. The reviewers have never written about the topic mismatch with scope of the IJMS journal, as evidenced by the citations of our recent works.
Reviewer 3 Report
Even though this manuscript presents a set of relevant data, the quality and depth of the discussion should be improved, specially concerning mechanical properties. Some specific comments:
Natural fibers establish hydrogen bonds with ENR, which may indeed explain the decrease in swelling when these are added. However, the sentence “natural fibers addition might contribute to the improvement in the density of crosslinks created during the rubber vulcanization process.” is confusing, since vulcanization creates covalent (chemical) crosslinking, while hydrogen bonds are physical crosslinks. I suggest that this difference be elucidated in the text, clarifying that both chemical and physical crosslinking is present in this system.
Swelling may also be affected by the filler’s affinity towards the solvent, in addition to degree of crosslinking. Authors should consider whether this should be discussed when analyzing, for instance, the lower swelling exhibited by ENR/PLA + CF or FF.
The sentence “the results of mechanical tests presented in Fig. 2b-c are consistent 161 with the previously detected swelling behaviour” does not portray the reality. The swelling of ENR/PLA + CF is equivalent to that of ENR/PLA + FF and significantly lower than for unfilled ENR/PLA, but this does not correlate with the tensile strength values!
The interpretation of the highest tensile strength provided by FF in comparison to CF is not very convincing. Cellulose is highly capable of establishing hydrogen bonds, despite being a “pure” material.
Also the interpretation of the results when MMT is added should be discussed in more detail. The conjugated effect of these fiber and clay fillers is that elongation at break increases while everything else remains essentially the same in relation to the unfilled material. The explanation provided is not coherent. For instance, a “reinforcing effect” of MMT would not lead to higher elongation, but to higher tensile strength.
Biodegradation should be quantified in terms of mass loss, not just visual inspection. Otherwise, only microorganism growth on the surface is being analyzed, and not the actual extent of material degradation.
It is not clear how the reported effects of these fillers might actually be relevant for making ENR/PLA competitive in relation to conventional synthetic polymers, as the authors claim, since no basis of comparison is provided. What specific polymer(s) are we trying to replace? What is the target performance? What is the filler(s) that better contributes to approaching that target?
English language should be improved throughout the text, in order to remove some mistakes and awkward expressions.
Author Response
Institute of Polymer and Dye Technology
Lodz University of Technology
90-924 Lodz, ul. Stefanowskiego 12/16, Poland
Tel.: +48 42 631 32 93, Fax: +48 42 636 25 43
March 8, 2021
International Journal of Molecular Science MDPI
Dear Editor,
I am sending you our manuscript entitled “Biocomposites of epoxidized natural rubber/ polylactic acid (ENR/PLA) modified by natural filler” by Anna Masek, Stefan Cichosz and Malgorzata Piotrowska with a request to publish the paper in International Journal of Molecular Science MDPI. The manuscript has not been previously published, is not currently submitted for review to any other journal, and will not be submitted elsewhere before a decision is made by this journal. All authors have approved the manuscript and agree with its submission to International Journal of Molecular Science MDPI.
For correspondence please use the following information: corresponding author: Anna Masek
Institute of Polymer and Dye Technology Technical University of Lodz
90-924 Lodz, ul. Stefanowskiego 12/16, Poland
Tel.: +48 42 631 32 13
Fax.: +48 42 636 25 43
e-mail: anna.masek@p.lodz.pl
PhD, DSc, Anna Masek
Associate Professor
Technical University of Lodz, Institute of Polymer and Dye Technology,
Stefanowskiego 12/16, 90-924 Lodz, Poland
Reviewer #3
Even though this manuscript presents a set of relevant data, the quality and depth of the discussion should be improved, specially concerning mechanical properties. Some specific comments:
The comments are listed below:
- Natural fibers establish hydrogen bonds with ENR, which may indeed explain the decrease in swelling when these are added. However, the sentence “natural fibers addition might contribute to the improvement in the density of crosslinks created during the rubber vulcanization process.” is confusing, since vulcanization creates covalent (chemical) crosslinking, while hydrogen bonds are physical crosslinks. I suggest that this difference be elucidated in the text, clarifying that both chemical and physical crosslinking is present in this system.
Answer: We are thankful for drawing our attention to this problem. The mentioned above phenomenon has been described more precisely: Thus, it could be concluded that natural fibers (CF, FF) addition might contribute to the improvement in the density filler-polymer matrix interactions. This may be caused by both the physical interactions between hydroxyl groups of cellulosic material and oxirane rings of ENR or ester moieties of PLA (Fig. 1) and covalent chemical bonds created during the vulcanization process. Thus, both physical and chemical crosslinking is present in the system described above.
- Swelling may also be affected by the filler’s affinity towards the solvent, in addition to degree of crosslinking. Authors should consider whether this should be discussed when analyzing, for instance, the lower swelling exhibited by ENR/PLA + CF or FF.
Answer: We are grateful for this advice, as this problem is worth considering. Therefore, the following sentence has been added to the text: Moreover, lower swelling of CF- or FF-loaded specimens may be also affected by the filler’s poor affinity towards the solvent, herein toluene.
- The sentence “the results of mechanical tests presented in Fig. 2b-c are consistent 161 with the previously detected swelling behaviour” does not portray the reality. The swelling of ENR/PLA + CF is equivalent to that of ENR/PLA + FF and significantly lower than for unfilled ENR/PLA, but this does not correlate with the tensile strength values!
Answer: We are sorry for this expression which was a shorthand. This sentence has been removed from the manuscript.
- The interpretation of the highest tensile strength provided by FF in comparison to CF is not very convincing. Cellulose is highly capable of establishing hydrogen bonds, despite being a “pure” material.
Answer: We are thankful for this comment. The discussion has been improved: Flax fibers compose of various additional substances rich in hydroxyl moieties, e.g., lig-nin, hemicellulose, which may affect the polymer matrix-filler interface and intensify the predicted interactions (Fig. 1). On the other hand, cellulose is a material with a homoge-nous chemical composition, simultaneously, being capable of establishing hydrogen bonds. Despite some similarities, cellulose and flax fibres may significantly differ consid-ering their surface characteristics, e.g., hydrophilicity, specific surface area, accessibility of surface functional groups, which may affect the filler’s behaviour in the polymer matrix and its effect on the properties of the polymer blend [43].
- Also the interpretation of the results when MMT is added should be discussed in more detail. The conjugated effect of these fiber and clay fillers is that elongation at break increases while everything else remains essentially the same in relation to the unfilled material. The explanation provided is not coherent. For instance, a “reinforcing effect” of MMT would not lead to higher elongation, but to higher tensile strength.
Answer: We have explained the differences between the samples when MMT is added and provided some examples from literature with similar results and explanations covering the following aspects: particle shape, size and alignment, dispersion, effect on PLA crystallization, impact on crosslinking. Discussion part has been improved as follows: Moving forward, sample filled with both FF and MMT exhibits lower tensile strength which is on the level of ENR/PLA + CF specimen. Nevertheless, ENR/PLA + FF + MMT sample reveals the highest elongation at break of approximately 600%. This phenomenon could be explained with the synergic effect of MMT particles alignment within the structure of the polymer matrix (MMT plates are high aspect ratio particles; their orientation is a significant factor regarding the mechanical performance) and potentially lower cross-linking density (evidenced before) which lead to the material plasticization [46,47]. Similar effect has been observed in the work presented by Keawkumay et al. [48]. According to the results presented in the research, a certain MMT surface treatment may contribute to the lowering of rubber’s cross-linking density and, thus, decrease in the mechanical performance of a composite. Moreover, Wang et al. [49] underline the importance of an appropriate filler dispersion within the polymer matrix. The authors present results for MMT contents from 1-5 wt%. The significant increase in the composite tensile strength is observed only in case of low MMT contents. On the other hand, Jiang et al. [50] draw an attention to the problem of high spe-cific surface area and shape of MMT particles – with their large L/D ratio, the MMT platelets induced lower stress concentration, which partly contributed to the higher elongation of MMT-filled blend. Similarly, scientists report that the highest values of tensile strength and elongation at break are evidenced for the MMT content up to 2.5 wt%. This indicates that a specific aluminosilicate amount is required in order to ensure the most satisfying mechanical properties of a final product. Additionally, it was proven by Papageorgiou et al. [51] and Mathew et al. [52] that natural additives such as MMT and cellulose might significantly influence the crystallization behaviour contributing to some variations in composite mechanical proper-ties. According to the researches, these fillers may behave as nucleating factors and promote the crystalline domains formation. Yet, it was reported that MMT incorporation may result in the considerable imperfectness of created crystals and, thus, it might contribute to the lowering of the stiffness and performance of a PLA-containing composite.
- Biodegradation should be quantified in terms of mass loss, not just visual inspection. Otherwise, only microorganism growth on the surface is being analyzed, and not the actual extent of material degradation
Answer: these were model conditions and such full biodegradation studies, eg in natural conditions in soil, will be done later.
- It is not clear how the reported effects of these fillers might actually be relevant for making ENR/PLA competitive in relation to conventional synthetic polymers, as the authors claim, since no basis of comparison is provided. What specific polymer(s) are we trying to replace? What is the target performance? What is the filler(s) that better contributes to approaching that target?
Answer: The aim of the study was to prepare sustainable and easily degradable elastic blends of epoxidized natural rubber with poly(lactic acid) that were reinforced with flax fiber and montmorillonite. We hope that improved Conclusions section addresses all the problems mentioned by the Reviewer and draws attention to the idea behind the presented research: Concluding, the elastic blends of epoxidized natural rubber (ENR) with poly(lactic acid) (PLA) have been successfully created. Regarding the fact that they were rein-forced with flax fiber (FF) and montmorillonite (MMT), some additional functional properties were introduced, e.g., improved biodegradation potential, improved mechanical performance/elongation. Thanks to the carried out research it was shown that FF incorporation into ENR/PLA blend may cause a significant improvement in tensile strength up to (19 ± 2) MPa, simultaneously contributing to the loss of an ability of a polymer blend to elongate (material stiffening). Contrary, it was presented that MMT employment in the role of the filler might lead to ENR/PLA blend plasticization and enables polymer blend’s elongation on the level of approximately 600%. Nevertheless, MMT-filled material exhibits significantly lower tensile strength in comparison with the samples filled with only plant-derived fibers. Moreover, based on the gathered results it can be concluded that the composite samples subjected to elevated temperature and UV irradiation undergo some notable changes in their properties. It was established that MMT incorporation may delay the degradation of the ENR/PLA elastic blends under mentioned above conditions without affecting their biodeterioration potential. Thanks to the carried out mould fungi tests, it was proven that plant-derived fibers incorporated into ENR/PLA blends might significantly enhance their biodegradation potential enabling faster and more efficient growth of the microorganisms. Importantly, the biodeterioration ability is not affected by the MMT incorporation. Significantly, it should be mentioned that, in comparison with literature data, ENR/PLA blends prepared in this research not only did exhibit the improved degradation potential but also were characterized with the performance higher than noted for neat ENR vulcanizates. According to the article by Intharapat et al. [59], unfilled ENR products reveal a tensile strength of approximately 2 MPa. Additionally, while ENR filled with rice husk ash, the vulcanizate’s performance might be elevated up to 16 MPa [60] which is lower than the tensile strength of FF-filled ENR/PLA blend (ap-proximately 19 MPa). Obviously, the ENR/PLA blends’ ability to elongate is limited by the PLA content and, thus, the elongation at break of neat ENR vulcanizates is usually higher. Gathering the presented information, it can be stated that biodegradable and elastic ENR/PLA blends are good materials for producing consumer products characterized with a short either long lifespan depending on the organic-inorganic phase ratio which can be controlled with amounts of different natural additives. Furthermore, the pro-ecological character of the produced polymer blends and their mechanical performance make them competitive and enable their widespread application in various branches of industry, while meeting current ecological requirements.
Round 2
Reviewer 1 Report
Dear Authors
I read the reply to my comments and the new version of the manuscript. I understand that in this moment you are unable to perform further characterizations due to the lack of research material and funds. Even though a few of my comments have not been addressed directly, I find that the answers to the referee#3’s comments are helpful to close the concern that I raised. I appreciate the new discussion sections included in the paper and for this reason I think that the paper can be published in this form.
Reviewer 2 Report
I maintain my earlier opinion. If the authors do not agree, Please should indicate where in the article are presented the results in the field on molecular biology and molecular medicine.
Reviewer 3 Report
Thank you for your replies. I have no further comments.